

# A kingdom in decline: Holocene range contraction of the lion (*Panthera leo*) modelled with global environmental stratification

David M. Cooper[1,3], Andrew J. Dugmore[1,2], Andrew C. Kitchener[1,3], Marc J. Metzger[1] and Antonio Trabucco[4]

[1] Institute of Geography, University of Edinburgh, School of Geosciences,, Edinburgh, United Kingdom
[2] Human Ecodynamics Research Center and Doctoral Program in Anthropology, City University of New York (CUNY), NY, United States of America
[3] Department of Natural Sciences, National Museums Scotland, Edinburgh, United Kingdom
[4] Euro-Mediterranean Center on Climate Change, IAFES Division, Sassari, Italy

## ABSTRACT

**Aim**. We use ecological niche models and environmental stratification of palaeoclimate to reconstruct the changing range of the lion (*Panthera leo*) during the late Pleistocene and Holocene.

**Location**. The modern (early 21st century) range of the lion extends from southern Africa to the western Indian Subcontinent, yet through the 20th century this range has been drastically reduced in extent and become increasingly fragmented as a result of human impacts.

**Methods**. We use Global Environmental Stratification with MaxEnt ecological niche models to map environmental suitability of the lion under current and palaeoclimatic scenarios. By examining modelled lion range in terms of categorical environmental strata, we characterise suitable bioclimatic conditions for the lion in a descriptive manner.

**Results**. We find that lion habitat suitability has reduced throughout the Holocene, controlled by pluvial/interpluvial cycles. The aridification of the Sahara 6ka dramatically reduced lion range throughout North Africa. The association of Saharan aridification with the development of pastoralism and the growth of sedentary communities, who practised animal husbandry, would have placed additional and lasting anthropogenic pressures on the lion.

**Main Conclusions**. This research highlights the need to integrate the full effects of the fluctuating vegetation and desiccation of the Sahara into palaeoclimatic models, and provides a starting point for further continental-scale analyses of shifting faunal ranges through North Africa and the Near East during the Holocene. This scale of ecological niche modelling does not explain the current pattern of genetic variation in the lion, and we conclude that narrow but substantial physical barriers, such as rivers, have likely played a major role in population vicariance throughout the Late Pleistocene.

Corresponding author
David M. Cooper,
d.cooper@nms.ac.uk

## INTRODUCTION

The overall aim of this paper is to model the range changes of the lion (*Panthera leo*) driven by large-scale climate changes since the Last Glacial Maximum, and evaluate their likely consequences on population distribution and connectivity. This is essential contextualisation for current and continuing anthropogenic impacts on the species. The known historical range of the lion included much of Africa and southeastern Europe; it extended to the Near East, the Arabian Peninsula, and southwest Asia as far as the Indian Subcontinent (*Ray, Hunter & Zigouris, 2005*; *Schnitzler & Hermann, 2019*; *Yamaguchi et al., 2004*), but today this range is considerably reduced. The lion is an iconic symbol of both Africa and India, but is suffering from rapidly declining numbers and geographical range mostly due to human activities (*Bauer et al., 2016*). Key threats to the lion in both modern and historic times are habitat reduction, depletion of the wild prey base and direct persecution, which are frequently associated with livestock husbandry and management (*Bauer, De Iongh & Sogbohossou, 2010*; *Black et al., 2013*; *Inskip & Zimmermann, 2009*). The population and range contraction of the lion is even more pronounced, if the closely related taxa, *P. (l.) spelaeus* (Eurasian cave lion) and *P. (l.) atrox*, (American lion) are included and longer periods of time are considered.

Recent molecular studies recognise a deep genetic division between the 'northern' lions (West Africa, Central Africa and North Africa/Asia), and 'southern' lions (North East Africa, East/Southern Africa and South West Africa) (*Barnett et al., 2014*; *Bertola et al., 2016*). Population divergence between the northern and southern groups, recognised as subspecies *Panthera leo leo* and *P l. melanochaita* respectively, likely emerged since the last interglacial (120–140 ka) (*Bertola et al., 2016*; *De Manuel et al., 2020*). Similar patterns are proposed for other savanna megafauna in Africa (*Bertola et al., 2016*; *Lorenzen, Heller & Siegismund, 2012*), suggesting a common environmental driver for genetic and population differentiation.

Whilst there is agreement on evidence for long-term genetic splits across the historical range of the lion, there is considerable variance in the proposed timing of divergence between lion populations as expressed both by the differences between studies, and through the credible confidence intervals stated in each analysis. The proposed causes of long-term genetic differentiation between populations are the bioclimatic conditions associated with pluvial (wetter) and interpluvial (drier) conditions of the Late Pleistocene, which caused widespread changes to preferred habitat, and affected the efficacy of potential geographical barriers, such as large rivers (*Bertola et al., 2016*; *Lorenzen, Heller & Siegismund, 2012*). Similarly, the wider dispersal of the lion outside Africa has been attributed to changes in climate, with pluvial conditions in northern Africa and the Middle East around 60-47ka (*Timmermann & Friedrich, 2016*) being thought to have enabled lion range expansion across Eurasia (*Yamaguchi et al., 2004*). Whilst lions are known to cross rivers, increasing water levels of tributaries of the Okavango River/Delta have been shown to affect crossing frequency (*Cozzi et al., 2013*), and it is possible that the very large rivers of Africa have provided effective environmental barriers to lion dispersal, especially during the wetter conditions experienced in the Late Pleistocene. Conversely, ribbons of vegetation along
river systems may have acted as pathways for dispersal and/or connection through arid areas by providing corridors of favourable habitat for both lions and their prey. However, confidence intervals on genetic divergence times are typically wide (*Antunes et al., 2008*; *Barnett et al., 2014*; *Bertola et al., 2016*), and thus direct correlations between them and known bioclimatic changes lack certainty.

Over geological timescales a species' changing range is a key dimension to interpreting its evolutionary history. It is important to assess the likely drivers of shifting geographical ranges, establish scales, directions and rates of change, and to examine currently fragmented, and recently extirpated populations. Expanding and contracting range shifts may have occurred through climatic and geographical changes, human influences or to changes in species assemblages. The ecological niches of large mammalian carnivores at continental scales are largely dependent on climate (*Geffen, Anderson & Wayne, 2004*; *Varela et al., 2010*), which has been used to model the ranges of big cats across Africa and Eurasia (*Cooper et al., 2016*; *Li et al., 2016*; *Townsend Peterson et al., 2014*). Mammal species are likely to have tracked consistent climatic conditions since the LGM (*Martínez-Meyer, Peterson & Hargrove, 2004*) and palaeoclimatic data are commonly used to infer mammal range shifts from previous glacial conditions to the present (*Cooper et al., 2016*; *Kohli et al., 2014*; *Li et al., 2016*; *Rebelo et al., 2012*; *Varela et al., 2010*). The climatic conditions of the Last Glacial Maximum ($\sim$21 ka), mid-Holocene ($\sim$6 ka) and present day capture the climatic extremes of the Late Pleistocene, and therefore encapsulate the variable degrees of contiguity and vicariance between populations over this period.

Lions have a broad habitat tolerance, with optimal habitat comprising of moist open woodland and thick bush, scrub and grass savanna complexes, yet they are also able to survive in more arid environments (*Celesia et al., 2010*; *Eloff, 1973*; *Nowell & Jackson, 1996*). During the Pleistocene the combined mid-to-low latitude distribution of lions in general was almost ubiquitous except for hyper-arid desert and dense tropical rainforests (*Bertola et al., 2016*; *Nowell & Jackson, 1996*; *Yamaguchi et al., 2004*). However, the extent of favourable habitat has varied through the Quaternary in response to climate change (*Nowell & Jackson, 1996*; *Riggio et al., 2013*). By associating the environmental tolerances of the modern lion with palaeoclimatic data, we can establish the role of a key driver of change. Modelling based on these data can provide likely scenarios for the timeframes of population separation or connection through periods of turbulent climatic conditions, and give critical contextualisation to future threats and conservation management of this Vulnerable IUCN Red List species (*Bauer et al., 2016*).

In this paper we use modelling to explicitly address previous biogeographical speculation of population connectivity and dispersal driven by climate change. We assess likely scales of climate-driven changes by modelling suitable lion habitat for key periods that exhibit the extremes of bioclimatic conditions within the Late Pleistocene and Holocene (*Chevalier, Brewer & Chase, 2017*). We thereby constrain our understanding of shifting lion ranges and lion population contiguity through this period within clearly defined limits. We assume that the environmental niche of the lion has been stable. The flexible social structure and ecological niche of the lion is influenced by prey preferences and availability, and resource selection (*Bauer, de Iongh & Di Silvestre, 2003*; *Meena, 2009*). Fundamental changes to

lion social structure, beyond the limits of present day populations, are unlikely to have occurred over the time periods modelled within this study (*Yamaguchi et al., 2004*), giving us additional confidence in our niche modelling approach. Within Africa, we do not consider the role of any extant species in competitively excluding the lion from any otherwise bioclimatically suitable area (see *Comley et al., 2020*), and no other large African mammalian carnivore has become extinct during the Late Pleistocene or Holocene (*Faith, 2014*). We limit our analysis to modern lions, and exclude consideration of the extinct Eurasian cave lion because phylogenetic analysis shows that it diverged from modern lions ∼500 ka and the two lineages likely did not hybridise following their divergence (*De Manuel et al., 2020*).

This paper provides a new deep-time perspective on the current deteriorating state of lion populations. This is important for understanding the historical context of the species' present vulnerability, which could be further exacerbated by future global change.

## METHODS

### Approach

We use Global Environmental Stratification (*Metzger et al., 2013a*) with ecological niche models to map environmental suitability of the lion under current and palaeoclimatic scenarios. A Maximum Entropy (MaxEnt) ecological niche modelling approach is used in conjunction with the production of Global Environmental Stratification Strata/Zones (*Metzger et al., 2013a*; *Metzger et al., 2013b*; *Soteriades et al., 2017*) to explore the extent of lion distributions through the Late Pleistocene and Holocene under different climatic scenarios. Interpreting ecological niche modelling projections of novel climates can be challenging due to the complexity of considering the combined effects of multiple continuous input variables. However, outputs from niche model habitat suitability maps are not necessarily intuitive or meaningful to the end user (*Baldwin, 2009*; *Merow, Smith & Silander, 2013*). By examining modelled lion range in terms of Global Environmental Stratification (GEnS) (*Metzger et al., 2013a*), we characterise the bioclimatic niche of the lion in a descriptive manner. The methods described here are presented graphically within Fig. 1

### Data

Lion locality records were collected from across the known historical range, and from a range of independent sources (Table 1), so as to reduce the influence of sampling bias in our data (*Fei & Yu, 2016*). The temporal range of these records captures the climatic conditions experienced by modern lions and allows the inclusion of records from now extinct populations. The recognition of spatial error in locality data is an important consideration (*Raxworthy et al., 2007*). The maximum locality error of data used here is 50 km, which is acceptable given a ∼20 km diameter of lion home ranges, and much greater dispersal distance of male lions (*Funston et al., 2003*).

GEnS describes relatively similar biophysical environments, which are derived through statistical clustering of the principal components of four bioclimatic variables: Growing Degree-Days on a 0 °C base (*Metzger et al., 2013a*), Temperature Seasonality (*Hijmans et*

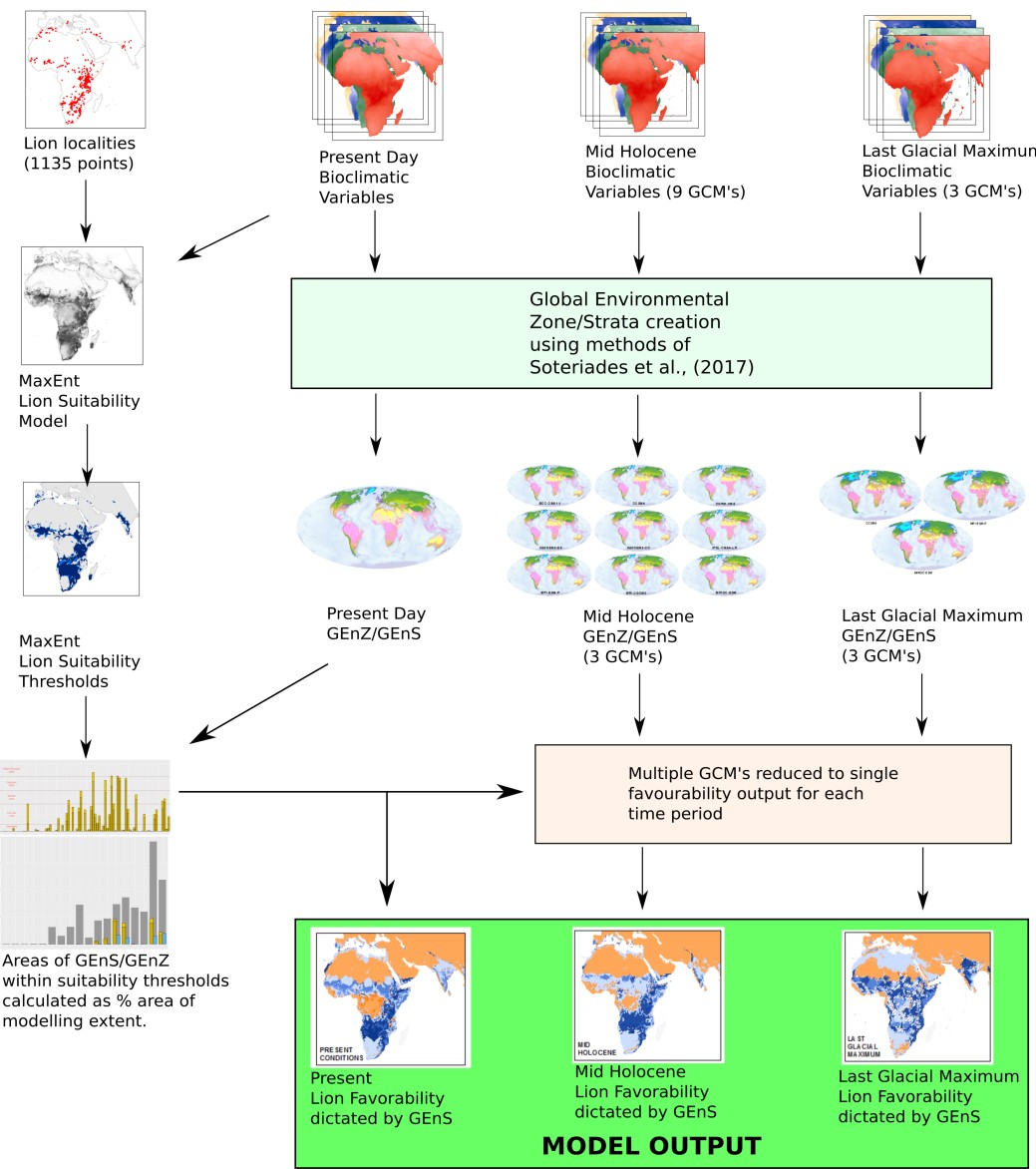

**Figure 1 Bioclimatic modelling methodology.** Bioclimatic variables from the present day, mid-Holocene and Last Glacial Maximum are clustered into Global Environmental Zones (GEnZ) and Strata (GEnS). The same bioclimatic variables are used in conjunction with lion localities to produce a MaxEnt lion suitability threshold model for the present day. Global Environmental Strata are then classified by suitability for the lion and displayed for each time period in terms of favourability. This method enables the production of habitat suitability maps within a descriptive bioclimatic framework.

*al., 2005*), Aridity Index and Potential Evapotranspiration Seasonality (*Zomer et al., 2007*; *Zomer et al., 2008*). Using the modelling approach of *Soteriades et al. (2017)*, the strata were created on a global scale for present day conditions, and for the mid-Holocene and the Last Glacial Maximum using coupled General Circulation Models (GCMs). We used a Random Forest classifier developed by (*Soteriades et al., 2017*) on each GCM to create 125
**Table 1  Source and number of locality points used in the ecological niche modelling process, compiled from new data (museum specimen locality descriptions) and from additional literature and resources.**

| Locality source | Number of localities |
|---|---|
| iNaturalist grade GBIF Localities (*iNaturalist, 2015*) | 781 |
| *Loveridge & Canney (2009)* | 134 |
| Museum Record Descriptions | 101 |
| VertNet Records (*Constable et al., 2010*) | 32 |
| *Black et al. (2013)* | 20 |
| *Barnett et al. (2014)* | 20 |
| *Banerjee & Jhala (2012)* | 12 |
| *Cross et al. (2009)* | 6 |
| **Total** | **1,135** |

multivariate strata characterised by similar climatic conditions. This was performed using the data-mining and machine-learning software Weka 3.6.4 (*Frank, Hall & Witten, 2016*) at a resolution. The strata were further aggregated into 18 easily interpretable, structured bioclimatic zones (*Metzger et al., 2013*). Global Environmental Zones are an established aggregation of Global Environmental Strata, created to provide consistent nomenclature and to support the summarising and reporting of results (*Metzger et al., 2013*). Global Environmental Zones (GEnZ) and Strata (GEnS) have been made available for present day, mid-Holocene and Last Glacial Maximum conditions at http://hdl.handle.net/10283/3274 (*Cooper et al., 2020*)—see Appendix S1.

## Ecological niche modelling

We used the four environmental variables of the GEnS classification (Growing Degree-Days on a 0 °C base, Temperature Seasonality, Aridity Index and Potential Evapotranspiration Seasonality) in our ecological niche model analysis to represent dominant bioclimatic trends. These four variables show the lowest correlation with each other and determine 99.9% of the total variation of 36 available bioclimatic variables (*Metzger et al., 2013*). The modelling extent (−19°W, 94°E, −36°S, 50°N) is defined by the area accessible to the lion over historical times.

The MaxEnt modelling approach was applied as outlined in *Cooper et al. (2016)* to create a habitat suitability model of the lion for the present day. The model was run to fit a Poisson point-process model by displaying raw output under the following settings: 'noremoveduplicatepresencerecords', 'noaddsamplestobackground'. Regularization multipliers of 2 and 100,000 background points were chosen (see Appendix S3 for full parameters). Model performance was measured using the mean area under the receiver operator curve (AUC) (*Phillips & Dudík, 2008*) from k-fold cross-validation and spatially independent cross-validation using the ENMeval package (*Muscarella et al., 2014*) in R (*R Core Team, 2015*). Spatially independent cross validation is important given the potential for spatial autocorrelation of our localities. Threshold values of suitable/unsuitable areas were derived from the MaxEnt model for comparison with global environmental strata.

A modified lowest-presence threshold (*Costa et al., 2010*) was used to determine a binary output of suitable lion habitat. We allowed an omission error of 10% ($e = 10\%$) to determine this threshold, which accounts for a level of uncertainty in the quality of our locality records (*Peterson, Papeş & Soberón, 2008*).

## Comparison of lion ecological niche model with global environmental strata

Climate stratification has been used in the creation of biological monitoring programmes, including the construction of sampling strategies for species distribution models (*Metzger et al., 2013*), but it has seen very limited application to the evaluation of these models or to mapping past faunal ranges (*Hickie, 2016*). Whilst the use of Global Environmental Stratification to create maps of suitability is visually similar to the underlying raw MaxEnt models, it also permits more in-depth analysis of preferred lion habitat, within a general descriptive framework that can be extended to other species, time periods and geographical locations.

The area of suitable habitat for the lion, dictated by the modelled threshold, was calculated for each environmental stratum and zone, as was the total extent used in modelling. Strata were categorised as highly favoured, favoured, utilised, low use and unsuitable, where modelled thresholds of suitable habitat account for 80–100% (highly favoured), 60–80% (favoured), 40–60% (utilised), 10–40% (low use) and <10% (unsuitable) of the total modelling extent. These categories were then expressed on the strata for the present day, mid-Holocene and Last Glacial Maximum scenarios. We display the modal value of suitability for multiple mid-Holocene and LGM models (or values, if two similar suitabilities cause a split agreement, e.g., favourable/highly favourable). If model results have no agreement, or the agreement is split between very different suitabilities, e.g., favourable/unsuitable, the strata were categorised as uncertain (see Appendix S2).

Additionally, the area of each Global Environmental Zone within the IUCN Red List's extant lion distribution data was calculated to compare the modelled fundamental niche with the realised niche of current lion range (Fig. 2). To provide environmental context to Global Environmental Zones and Strata within the modelling extent, the proportion of MODIS land cover classes represented by each Zone and Strata was calculated.

# RESULTS

## Lion environmental preferences

The quantified climatic preferences of the lion, using a MaxEnt ecological niche model of habitat suitability, are shown in Fig. 3. The AUC value from model 10-fold cross validation was 0.923. The model AUC score from spatially independent cross validation, using the 'checkerboard2' method (*Muscarella et al., 2014*), was 0.818. The modified lowest-presence threshold ($e = 10\%$), derived from the MaxEnt model, was used to calculate 'highly favoured', 'favoured', 'utilised', 'low use' and 'unsuitable' Global Environmental Strata and Zones within the modelling extent (Figs. 4 and 5). Highly favoured and favoured lion habitats predominantly consist of hot and mesic, hot and dry, extremely hot and xeric, and extremely hot and moist environmental zones (Fig. 4). The modelled scenarios

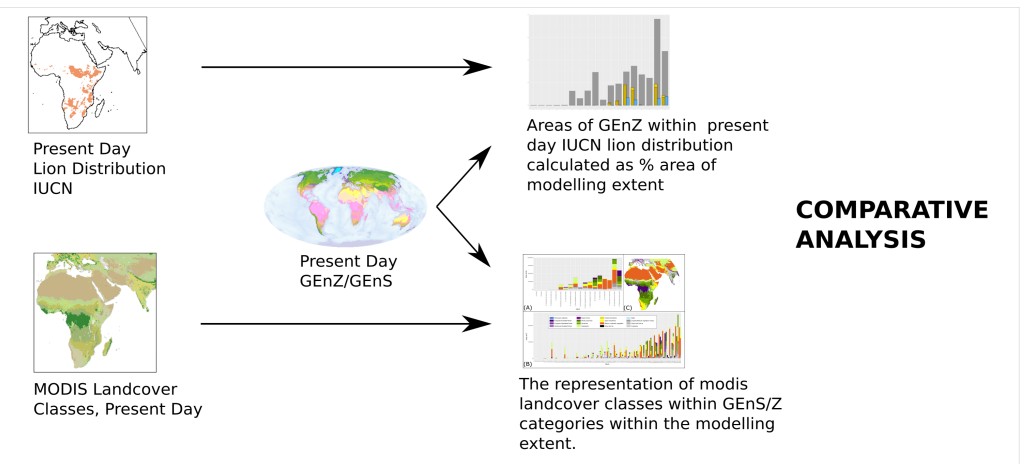

**Figure 2** **Methodology for the comparative assessment of favourable lion range with present day range and landcover.** Present day IUCN lion distribution is compared with Global Environmental Zones to visualise the current realised niche of the lion. MODIS Landcover Classes for the present day are compared with Global Environmental Strata within the modelling extent to provide landcover examples for each stratum.

show a wider present day habitat tolerance than current known lion distributions derived from IUCN data (*Bauer et al., 2016*), with some favourability modelled within warm temperate zones (Fig. 4). Within preferred environmental zones, some strata are low use or unsuitable. We compared each environmental stratum to MODIS (*Channan, Collins & Emanuel, 2014*; *Friedl et al., 2010*) land-cover classes to gain insight into the typical land covers of each stratum (see Appendix S6). Today's lions prefer strata which are typified by woody savannas and savannas, rather than barren/sparse cover or more closed forest covers, which can occur within the same broader environmental zones.

## Modelled Holocene environmental suitability for the lion

We projected preferred lion habitat upon modelled Global Environmental Strata for present day conditions, to the mid-Holocene ∼6 ka and Last Glacial Maximum (LGM) ∼21ka (Fig. 6). Mid-Holocene and LGM outputs were derived from multiple coupled GCMs from the Paleoclimate Modelling Intercomparison Project Phase III (PMIP3) and downscaled at five arc-minute resolution (see Appendix S4). Suitable conditions for the lion have fluctuated considerably since the LGM. Compared with present day interglacial/interpluvial conditions, the LGM was considerably more favourable for the lion in both overall 'favourable' environmental conditions and connectivity across the historical range. Favourable conditions across models have consistently been reduced for the Indian Subcontinent from the LGM through the mid-Holocene and into present day conditions. Figure 7 highlights the reduction of highly favoured, favoured and utilised strata and increases in unsuitable and low-use strata from the LGM to the present. Whilst LGM conditions are more favourable to the lion than modelled mid-Holocene or present day conditions, the core 'favourable' environment has shifted markedly. During the LGM, the favourable lion habitat is consistently modelled across the Sudanian region
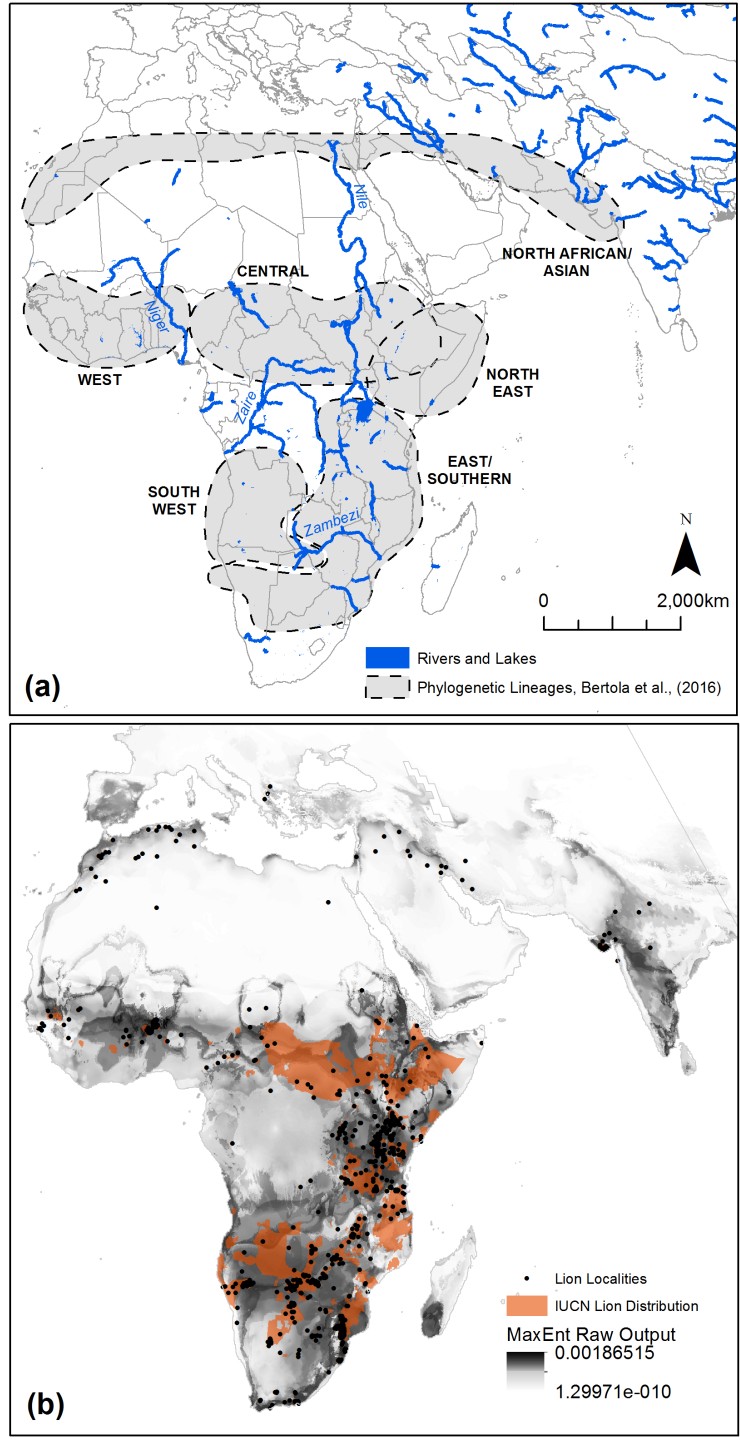

Figure 3 **MaxEnt modelling results, genetic demarcations of the modern lion, and potential biogeographical barriers within Africa and the Near East.** We highlight the proposed genetic demarcations of the modern lion across Africa, the Near East and southern Asia (*Bertola et al., 2016*) and the location of large rivers and lakes as potential influencers of lion dispersal that are not accounted for in our models (A). The raw output of our MaxEnt model displays areas climatically favourable to the lion (B). The current known range of the modern lion (orange) is restricted to a subset of modelled favourable habitat.

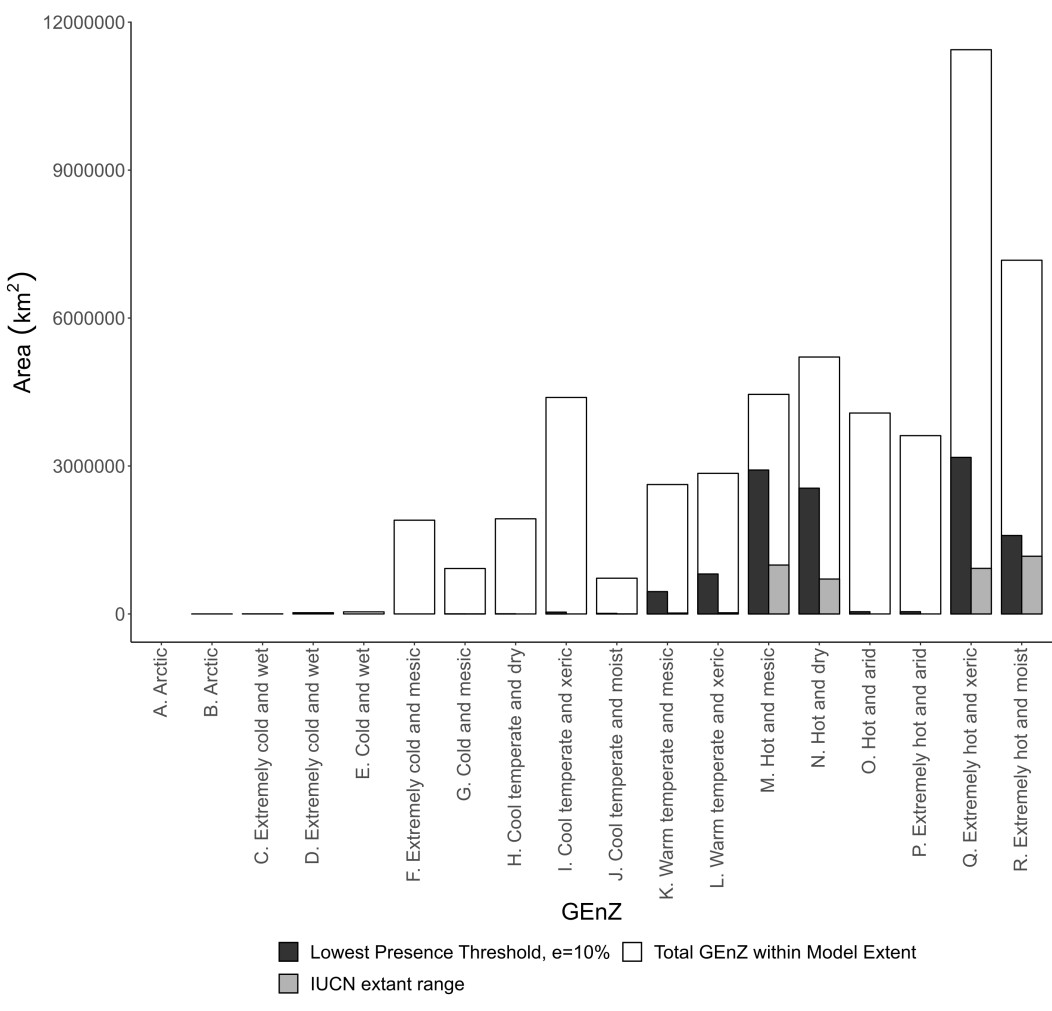

**Figure 4** **The proportions of Global Environmental Zones that are occupied by modelled lion distribution based on climatic suitability, and on IUCN extant lion range within our modelling extent.** The proportion of Global Environmental Zones that is occupied by modelled lion distribution based on climatic suitability, and on IUCN extant lion range within our modelling extent of Africa, the Near East and the Indian Subcontinent. Lions occupy warm temperate and mesic, hot and mesic, hot and dry, extremely hot and xeric, and extremely hot and moist habitats as shown by both modelled results and extant distributions. Hot and mesic, and hot and dry habitats are particularly favoured under idealised model scenarios. Significant reductions in extant range, compared with modelled range, were likely to be caused by anthropogenic pressures.

with comparatively less favourable conditions than the mid-Holocene and present day in southern Africa. The modelled results show the LGM as the most likely time for dispersal out of Africa to the Near East and Indian Subcontinent, but the extent and quality of the linkage is low. In all timeframes and all modelling scenarios there is little suitable habitat modelled within the Near East, and modelled suitability in southern Europe has a small geographical range. The Congo basin has progressively become less favourable to the lion since the LGM (Fig. 8).

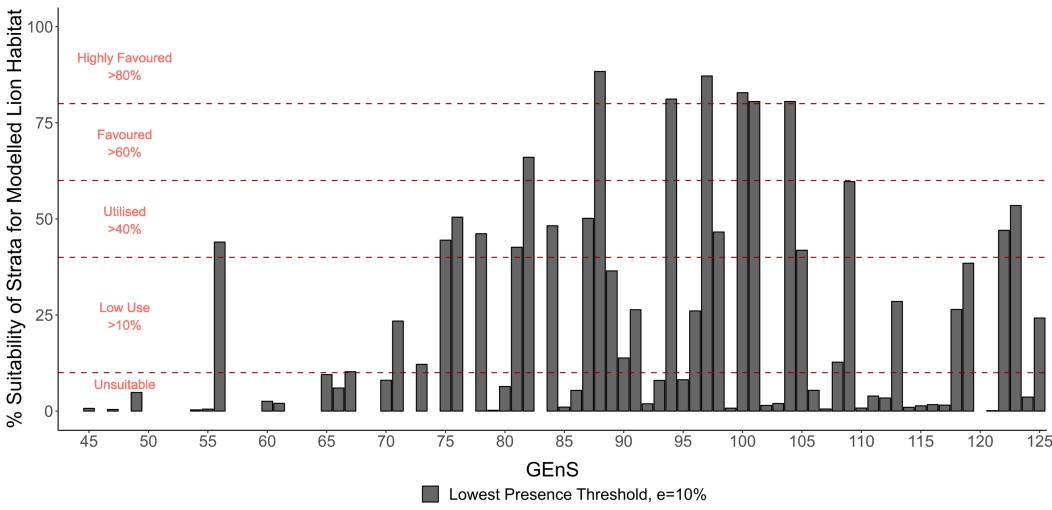

**Figure 5** **The percentage of Global Environmental Strata occupied by the modelled distribution of the lion within the modelling extent of Africa, the Near East, and the Indian Subcontinent.** This was used to inform maps of favourable Global Environmental Strata for the lion (Fig. 4).

## DISCUSSION

The mapping of lion habitat suitability, in terms of Global Environmental Strata, provides an insight into their preferred Global Environmental Zones. Whilst certain environmental zones are more favoured by lions, no zone is modelled as universally suitable for the lion, as both favoured and highly favoured strata are found within environmental zones that include unsuitable and low-use strata. This is probably due to the lions' wide habitat tolerance within transitional landscapes, but limited tolerance of climatic extremes. We expect lion distributions within semi-desert, but not true desert (Extremely hot and xeric), and in tropical forest, but not dense rainforest (Extremely hot and moist). The non-linear nature of vegetation cover through climatic gradients (*Scheffer et al., 2012*) has likely also played a role in the complex suitability of each environmental zone, as highlighted in the association between environmental strata and MODIS landcover (see Appendix S6).

Our modelling of the LGM shows some limited climatic suitability for lion dispersal between Africa and the Indian Subcontinent. This potential is most pronounced south of the present day An Nafud desert, through the northern Persian Gulf, and eastward through the southern Zagros Mountains and Balochistan. This corridor is characterised by warm temperate and xeric, warm temperate and mesic, and hot and dry environmental zones, and strong environmental gradients across the strata (Fig. 7). It has been assumed that the lion moved out of Africa via the Sinai Peninsula (*Barnett et al., 2014*), but the potential dispersal of Hamadryas baboons, *Papio hamadryas*, to Arabia via the Bab-el-Mandab during the Late Pleistocene (*Kopp et al., 2014*), when sea levels were lower, raises the possibility that this route was also used by the lion. Whilst lions may have crossed the narrow strait to small areas of favourable habitat, our analysis indicates a parallel and better supported dispersal could have occurred from the Sinai Peninsula into the Arabian Peninsula. Whilst

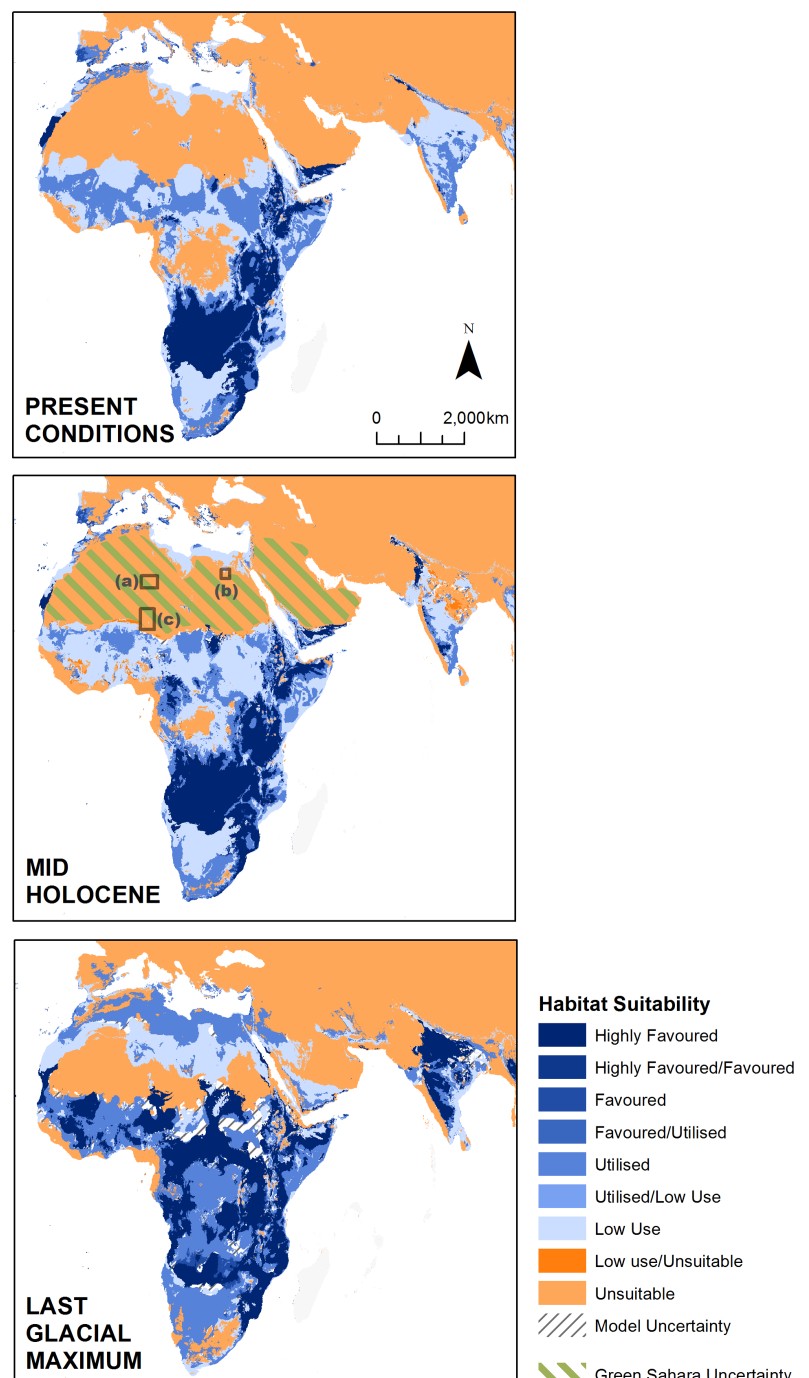

**Figure 6** **Modelled lion habitat suitability for the present day, mid-Holocene and Last Glacial Maximum, based on global environmental strata (GEnS).** Mid-Holocene and Last Glacial Maximum maps represent the combined suitability based upon nine and three coupled general circulation models respectively. We include an area of uncertainty surrounding the mid-Holocene greening of the Sahara and Arabia (*Hoelzmann et al., 1998*; *Larrasoaña, Roberts & Rohling, 2013*) and evidence of Lions and other savanna megafauna at (A) Tassili n'Ajer, (B) Wadi el-Obeid and (C) Aïr (*Galvin, 2018*) (see Appendix S5 for records).

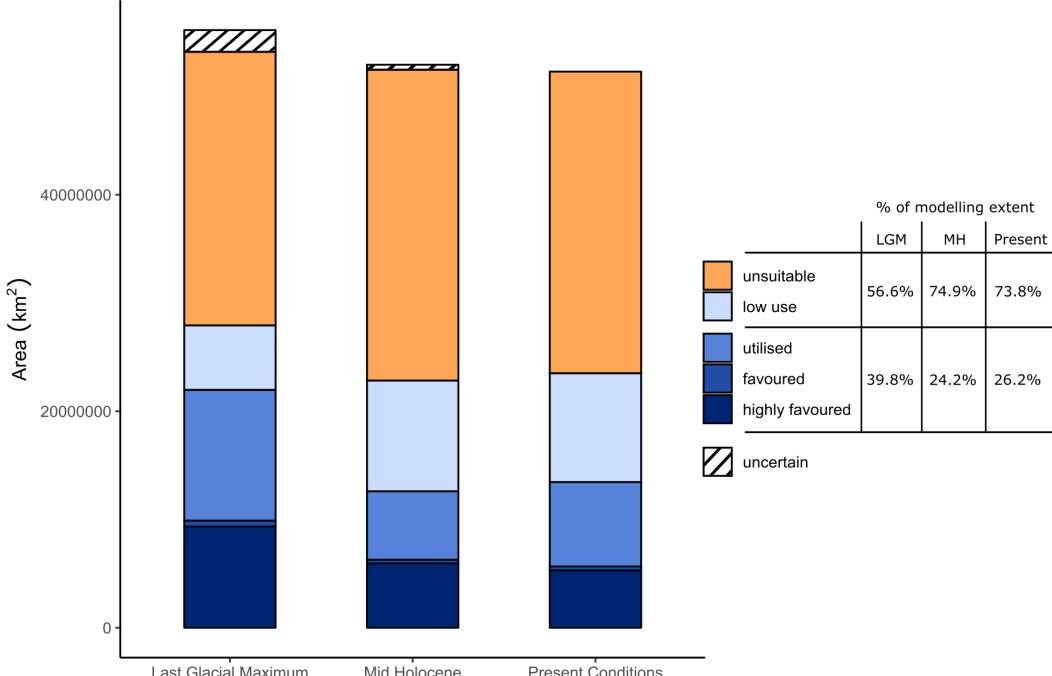

**Figure 7** **Changing area of favourable climatic conditions for the lion.** The changing area of favourable climatic conditions for the lion is shown for the present day, the mid-Holocene and the Last Glacial Maximum within Africa, the Near East and the Indian Subcontinent. Where model uncertainty exists between underlying mid-Holocene projections, we collapsed the classes "highly favoured/favoured", "favoured/utilised", "low use/utilised" and "unsuitable/utilised" into "highly favoured", "favoured", "utilised" and "low use" respectively. Utilised, favoured and highly favoured strata are more prevalent during the Last Glacial Maximum (39.8% of total area) than for either the Mid-Holocene (24.2%) or present day (26.2%), which are characterised by greater areas of unsuitable and low-use strata. The total area for the LGM is greater than the present day and mid-Holocene due to lower sea levels at this time.

arid conditions persisted within parts of the Sahara (*Adkins, DeMenocal & Eshel, 2006*), the more favourable conditions modelled for the lion within the Sahara/Near East during the LGM is consistent with widespread palaeoenvironmental records showing wetter conditions during this period (*Drake et al., 2011*; *Jennings et al., 2015*; *Larrasoaña, Roberts & Rohling, 2013*; *Migliore et al., 2013*).

The underlying GEnS/Z datasets we have created suggest that whilst increased vegetation may have penetrated desert zones up to 500 km northwards of today's southern limits, (as reported by *Willis et al. (2013)*), inhospitable, hot and arid, and extremely hot and arid climates persisted through much of the Sahara during the mid-Holocene (Fig. 9). Crucially, however, this persistence of hot and arid conditions is not consistent with a wide body of evidence suggesting that large parts of the Sahara were characterised by well-connected (mega) lakes, rivers and inland deltas during the African Humid Period ~11–4 ka (*Drake et al., 2011*; *Hoelzmann et al., 1998*; *Migliore et al., 2013*; *Tierney, Pausata & DeMenocal, 2017*; *Willis et al., 2013*), and populated by a diverse assemblage of present day sub-Saharan megafauna (*Yeakel et al., 2014*). In the absence of a comprehensive fossil record, we include the locations of African rock art which depict lions and other large megafauna, from within

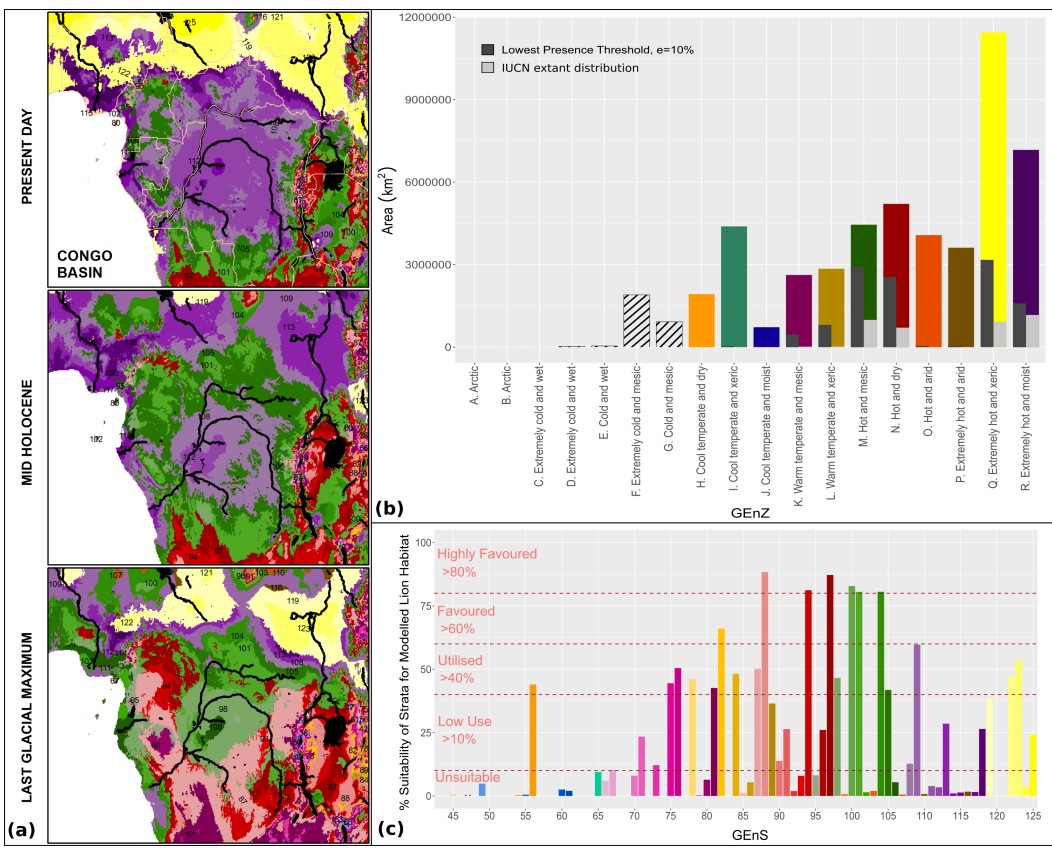

**Figure 8  Global Environmental Strata through the Holocene within the Congo Basin.** We display Global Environmental Strata (GEnS) for key areas across present, mid-Holocene and Last Glacial Maximum conditions within the Congo Basin (A). Strata colours are grouped into shades corresponding to Global Environmental Zone (B). Total area of each Global Environmental Zone within the modelling extent, and the suitability of each zone for the lion based upon our models, and on current lion range determined by the IUCN are displayed (B). The colours of strata (A) are matched to the associated bar chart of Global Environmental Strata (C), which displays the suitability of each stratum for the lion based upon the MaxEnt ecological niche model.

the African Humid Period ∼11-5ka (Fig. 6, see Appendix S5 for references). This supports a wider distribution of lions than suggested through our modelling. There is a strong argument that during the 'Green Sahara' episode, the Arabian Peninsula through to the western Indian Subcontinent also experienced wetter conditions, as these were affected by the same monsoonal forcing (*Hoelzmann et al., 1998*; *Jennings et al., 2015*; *Jones et al., 2013*). The disparity between GEnS/Z datasets and other palaeoenvironmental evidence exists because of the short-comings of the underlying palaeoclimatic data within the Saharo-Arabian Region which has driven our models. The PMIP3 experiments, which drive the WorldClim palaeoclimatic datasets in our models, do not reproduce the Green Sahara, because several driving mechanisms may not be accounted for in GCMs such as changes in desert dust, orbital changes and northward translation of the ITCZ, and vegetation feedbacks are either weak or non-existent (*Tierney, Pausata & DeMenocal,*

*2017*). As a result, our model is likely to have only captured a minimum distribution of lions within the Sahara during the mid-Holocene. In reality it is likely that during African Humid Period suitable lion habitat was far more extensive across the Sahara and Arabia, and probably southwest Asia as well, thereby providing stronger opportunities for dispersal and connections between populations. Individual lions are highly mobile, with individual males known to disperse >200 km within 1–2 years (*Funston et al., 2003*; *Van Hooft et al., 2018*), and even fleeting favourable conditions would have created opportunities for genetic flow. Given the overall weight of evidence, it is likely that during the African Humid Period a series of connected rivers, lakes and deltas existed across the Sahara (*Drake et al., 2011*) . The ecological changes related to this altered hydrology would have either facilitated the movement of lions across the region by creating both extensive savanna and favourable lakeshore and riparian habitat corridors, or constrained their dispersal by creating water barriers to movement. Thus, the recognition (and confirmation) of a 'Green Sahara' has significant implications for a more detailed understanding of long-term variations of lion population size, and patterns of dispersal within and out of North Africa and the Near East.

There is good evidence for the presence of lions within the Near East and southeastern Europe into historical times (*Bartosiewicz, 2009*; *Schnitzler, 2011*), yet our modelling shows that today these regions have a particularly poor climatic suitability for lions (Figs. 6 and 10). One explanation for this could be lion survival in inter-pluvial refugia formed around river systems and water points, and the endurance of relict populations from a previous contiguous range (*Black et al., 2013*). These limited climatic refugia would have made lion populations particularly vulnerable to anthropogenic pressures within the region, leading to their local extirpation within historical times.

We argue that recent historical populations in the Near East are not an indication of long-distance dispersal routes given the presence of climatic barriers and lack of continuous riparian corridors. About 6,000 years ago, the latest phase of aridification across the Sahara and Arabia probably separated lions in India from those in Africa. The Gir population appears to be a relict of more favourable palaeoclimatic conditions, but today there are still significant areas within the Indian Subcontinent which appear to be climatically favourable, such as the Deccan Plateau east of the Western Ghats. Within this eastern range, potential ecological competition with the sympatric tiger, *Panthera tigris*, may have constrained the lion's potential present day distribution.

Our modelling indicates that through the pluvial/interpluvial cycles of the Holocene the sub-Saharan range of lions has always been contiguous, and so we have no evidence for any significant gaps between populations caused by depopulated zones of climatically unfavourable terrestrial habitat. An analysis of nuclear DNA from historical samples of lions (*Curry et al., 2020*) supports contiguity amongst modern lion populations (but not amongst present day fragmented populations), but mitochondrial DNA (mtDNA) suggests regional subdivisions. This reflects sex-biased dispersal amongst lions, with males dispersing from natal prides while females remain resident. Sex-biased dispersal implies that something else has created barriers to female lion dispersal between Western African and Eastern/Southern African populations. This is significant given the pre-Holocene mtDNA divergence amongst these lion populations (*Antunes et al., 2008*; *Barnett et al.,*

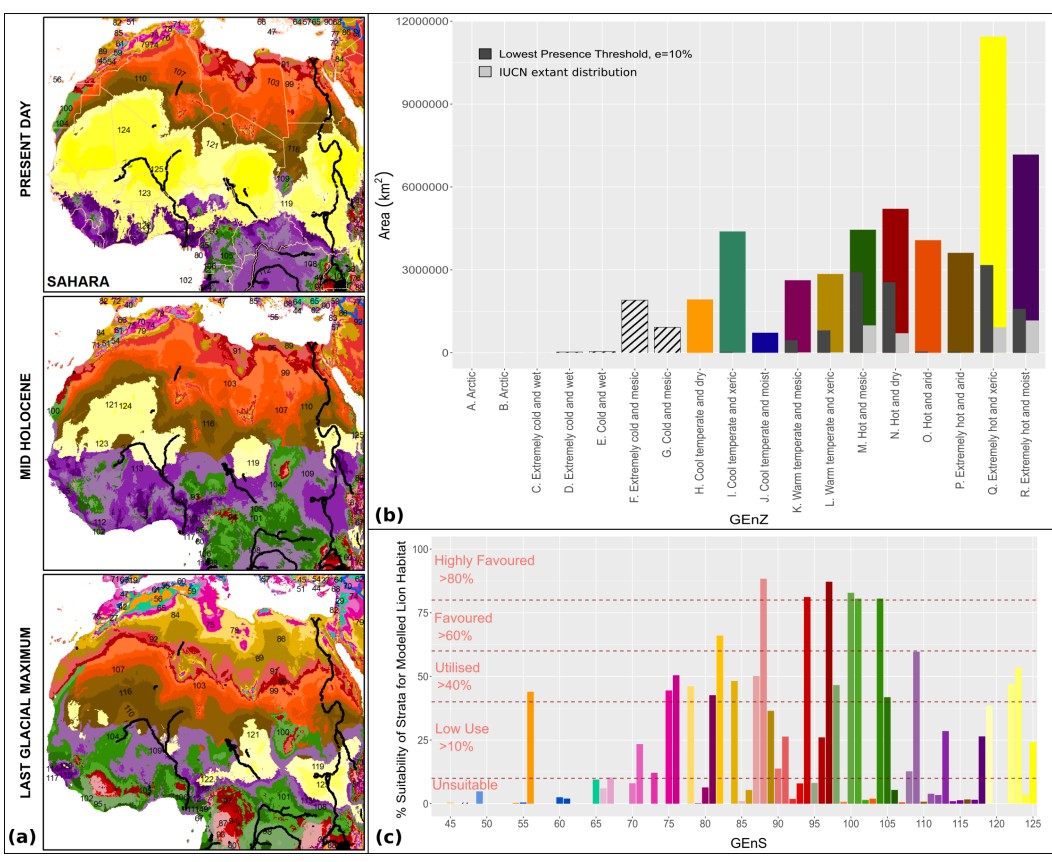

**Figure 9 Global Environmental Strata through the Holocene within the Sahara.** We display Global Environmental Strata (GEnS) for key areas across present, mid-Holocene and Last Glacial Maximum conditions within the Sahara(A). Strata colours are grouped into shades corresponding to Global Environmental Zone (B). Total area of each Global Environmental Zone within the modelling extent, and the suitability of each zone for the lion based upon our models, and on current lion range determined by the IUCN are displayed (B). The colours of strata (A) are matched to the associated bar chart of Global Environmental Strata (C), which displays the suitability of each stratum for the lion based upon the MaxEnt ecological niche model.

*2014*; *Bertola et al., 2016*; *De Manuel et al., 2020*; *Curry et al., 2020*), and numerous other large mammalian grassland/savanna species (*Bertola et al., 2016*). Thus, patterns of regional climate alone are not able to explain longer-term genetic divergence between populations. A combination of less favourable climatic conditions surrounding Lake Turkana within Africa's Rift Valley, and the presence of major physical barriers, such as Lake Turkana itself, other Rift Valley lakes, the Omo River and Nile River systems (Fig. 11), may have significantly reduced gene flow for large, mobile mammal species such as the lion. Further west, the Niger river system may have substantially reduced gene flow between central and West African populations. The contiguous favourable conditions found in eastern southern Africa throughout the changing climatic conditions of the Late Pleistocene supports the argument for this region as the evolutionary cradle of the modern lion (*Barnett et al., 2014*). Given the identified discrepancies between the coupled General Circulation Models and

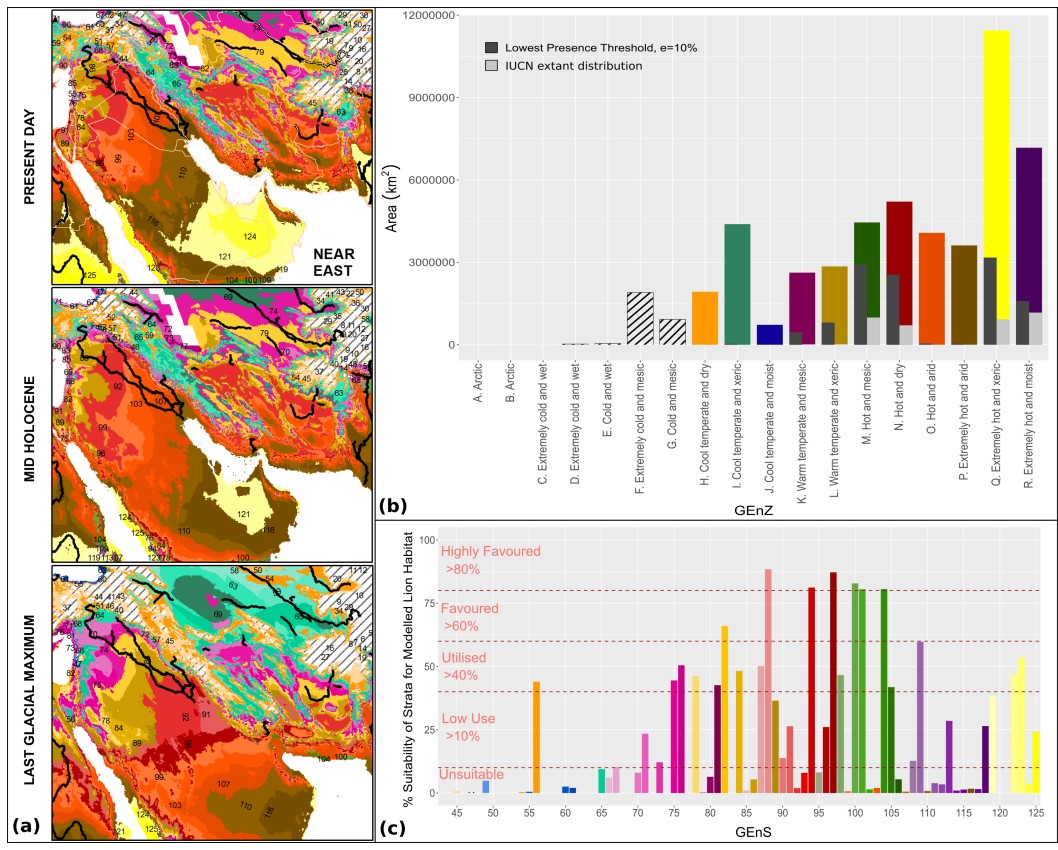

**Figure 10 Global Environmental Strata through the Holocene within the Near East.** We display Global Environmental Strata (GEnS) for key areas across present, mid-Holocene and Last Glacial Maximum conditions within the Near East (A). Strata colours are grouped into shades corresponding to Global Environmental Zone (B). Total area of each Global Environmental Zone within the modelling extent, and the suitability of each zone for the lion based upon our models, and on current lion range determined by the IUCN are displayed (B). The colours of strata (A) are matched to the associated bar chart of Global Environmental Strata (C) , which displays the suitability of each stratum for the lion based upon the MaxEnt ecological niche model.

climatic proxy data in the northern hemisphere, it is possible that model inconsistencies exist within other regions of interest. PMIP3 experiments show good agreement with palaeoclimate proxy data for the mid-Holocene within eastern Africa, and this region was likely wetter than is modelled during the LGM, whilst south Eastern Africa was likely drier and cooler (*Barker & Gasse, 2003*; *Chevalier, Brewer & Chase, 2017*), with African Rift Valley lake levels similar to those found today (*Barker & Gasse, 2003*). However, the impact of these differences during the LGM is not of the same scale or extent as those found within the Sahara and Arabia, during the mid-Holocene. The lowest agreements between PMIP3 models occur towards coastal regions, and differences are not homogenous across East Africa (*Singarayer & Burrough, 2015*). However, it is possible that different conditions during the LGM within eastern Africa could have affected lion distributions, and therefore could have contributed to the long-term vicariance of northern and southern populations.

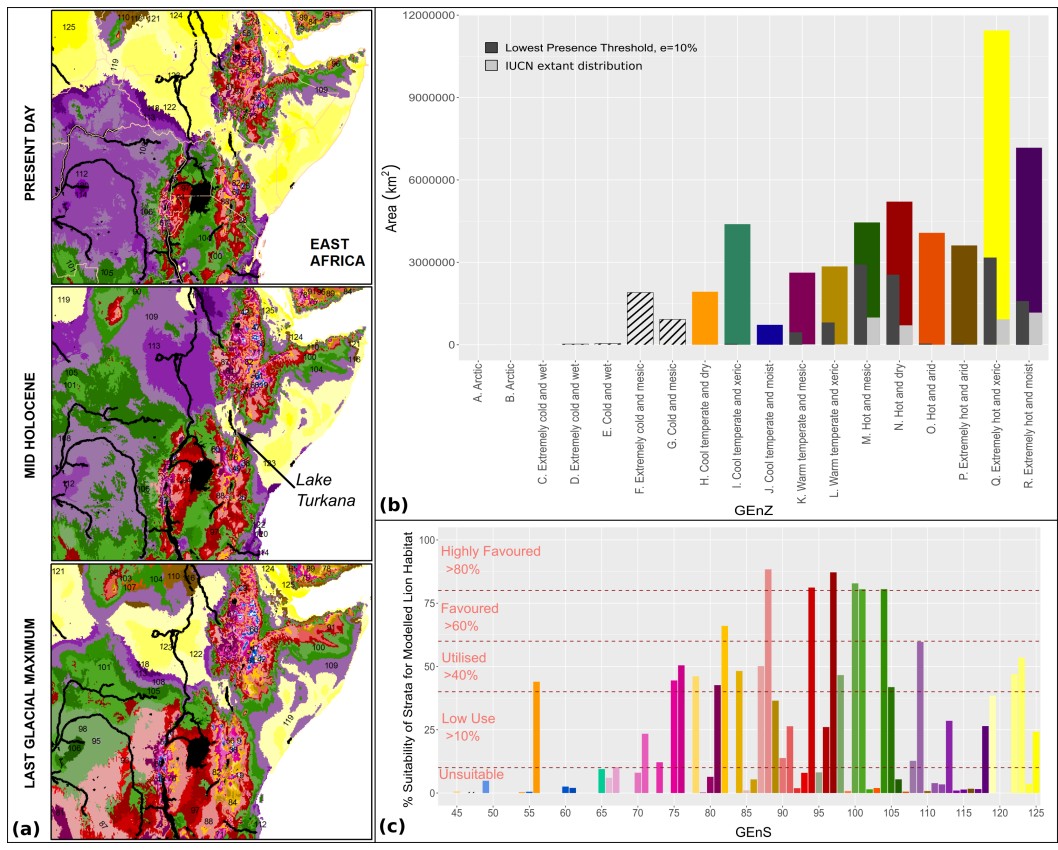

**Figure 11** **Global Environmental Strata through the Holocene within East Africa.** We display Global Environmental Strata (GEnS) for key areas across present, mid-Holocene and Last Glacial Maximum conditions within East Africa (A). Strata colours are grouped into shades corresponding to Global Environmental Zone (B). Total area of each Global Environmental Zone within the modelling extent, and the suitability of each zone for the lion based upon our models, and on current lion range determined by the IUCN are displayed (B). The colours of strata (A) are matched to the associated bar chart of Global Environmental Strata (C), which displays the suitability of each stratum for the lion based upon the MaxEnt ecological niche model.

The wetter conditions across North Africa might have effectively counteracted the range reduction in other areas since the LGM and would have allowed lion expansion from southern Africa to northern Africa. However, widespread mixing of populations is not supported by current genetic evidence except in northeast Africa, suggesting climate alone has not constrained lion population differentiation. In recent historical times the centre of gravity of lion distribution has lain in southern or eastern Africa, but in the past, it could have been much further north. Following the desertification of the Sahara, the range changes in the north were proportionally far greater than those in the south, and this may have led to increased genetic drift through population isolation (*Yamaguchi et al., 2004*). The reduction in suitable habitat for the lion through the Holocene, and especially following the aridification of the Sahara and Arabia has coincided with the development of agricultural systems and rising anthropogenic pressures, creating a double impact on

the lion. This modelled range change is consistent with previous skyline plots of lion populations, which show a recent precipitous drop (*Bertola et al., 2016*). The position of major rivers through Africa, in addition to contractions of suitable habitat driven by climate change, are the likely causes of vicariance over 100ka. Suture zones and parapatric (sub)speciation are likely important in maintaining genetic variation (*Bertola et al., 2016*; *Curry et al., 2020*; *Curry, White & Derr, 2019*; *Dures et al., 2020*). The persistence of major river barriers/corridors in the region, appears to have major biogeographical legacies in terms of defining boundaries between populations and linkage between areas.

The long-term trend in lion range reduction from the LGM into present day conditions as revealed by our modelling becomes even more pronounced if we consider the possible extent of more benign conditions across the Sahara and Arabia during the African Humid Period and the subsequent persistence of hyper-arid conditions through the region after ∼4.3 ka (*Kröpelin et al., 2008*). A rapid decline in overall lion numbers, as a result of mid-Holocene range contractions driven by climate change, is compatible with a population skyline plot derived from genetic analysis (*Bertola et al., 2016*). Although the ranges of lions south of the Sahara remained contiguous, climatic change may have led to poor connectivity amongst West African, North African, and European/Asian lion populations. In addition to climate change, the Holocene has witnessed increasing human impacts on lions because humans and lions flourish in the same areas (*Kuper & Kröpelin, 2006*), and ultimately competition for favoured habitats has driven the anthropogenic pressure on lions today. Conflicts are likely to have arisen alongside domestication and the development of pastoralism as lions would prove a significant threat as predators of cattle and other livestock (*Woodroffe, 2000*). Short-lived aridification of the Sahara ∼8 ka is associated with widespread transition to pastoralism from hunter-gathering (*Dunne et al., 2012*; *Tierney, Pausata & DeMenocal, 2017*). With increased aridification, human populations congregated with their domestic livestock within the same ecological refuges as lions (*Kuper & Kröpelin, 2006*), thus exacerbating direct conflict between people and lions that probably drove the local extinctions of lions, and created new barriers to lion dispersal and gene flow.

We conclude that there has been a long-term reduction of lion numbers during the Holocene, driven by the coincidental and combined influence of climate change and human impacts. The significant and continuing reduction in lion numbers during the 20th/21st centuries (*Bauer et al., 2016*; *Black et al., 2013*) is occurring in the context of a global population under increasing pressure. A mutually reinforcing effect of range reductions driven by climate and an intensification of human-lion conflicts, as found during the last ∼6ka, is likely to further intensify in the future.

## Core findings

I. Global Environmental Stratification provides a modelling framework that facilitates descriptive interrogation of our findings.

II. The presence of a Green Sahara/Arabia is not apparent in the Environmental Zones/Strata, as expected due to the shortfalls of PMIP3 simulations, and this limits our understanding of lion distributions in North Africa during the mid-Holocene.

III. Modelling does not identify any significant areas of favourable habitat for lions extending across the Zagros mountains or the Tigris-Euphrates river basin. As a result, we cannot identify any obvious climatic explanation for lion expansion out of Africa; although this is likely explained by poor model performance within North Africa and the Near East during the mid-Holocene.

IV. Modelling indicates that lion ranges south of the Sahara have been contiguous, so climatic drivers are not responsible for long-term vicariance in lion populations, which could instead be the result of the discrete geographical barriers formed by rivers, lakes, mountains, etc.

V. There has been a general reduction in lion range from the LGM, through the mid-Holocene to the present day.

## ACKNOWLEDGEMENTS

Thanks go to Andreas Soteriades and Jonny Hickie for support in utilising the Random Forests classifier within the Weka software.

### Funding

This work was funded under NERC PhD studentship NE/L002558/1 to David M. Cooper. The funders had no role in study design, data collection and analysis, decision to publish, or preparation of the manuscript.

### Grant Disclosures

The following grant information was disclosed by the authors:
NERC PhD studentship:  NE/L002558/1.

### Competing Interests

The authors declare there are no competing interests.

### Author Contributions

- David M. Cooper conceived and designed the experiments, performed the experiments, analyzed the data, prepared figures and/or tables, authored or reviewed drafts of the paper, and approved the final draft.
- Andrew J. Dugmore, Andrew C. Kitchener and Marc J. Metzger conceived and designed the experiments, authored or reviewed drafts of the paper, and approved the final draft.
- Antonio Trabucco analyzed the data, authored or reviewed drafts of the paper, and approved the final draft.

### Data Availability

The datasets generated and analysed during the current study are available in Edinburgh DataShare: Cooper, David M; Dugmore, Andrew J; Kitchener, Andrew C; Metzger, Marc J; Trabucco, Antonio. (2020). Lion range changes through the late Quaternary, modelled

using Global Environmental Stratification, [dataset]. The University of Edinburgh. School of Geosciences. Institute of Geography. https://doi.org/10.7488/ds/2509.

## Supplemental Information

Supplemental information for this article can be found online at http://dx.doi.org/10.7717/peerj.10504#supplemental-information.

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
