# Peer review of "A kingdom in decline: Holocene range contraction of the lion (Panthera leo) modelled with global environmental stratification"

_PeerJ, doi:10.7717/peerj.10504_

## Round 0.1 · original submission · Major Revisions

Many thanks for submitting what is a very interesting manuscript to PeerJ. It has now been received what I believe are two detailed reviews that will very much improve the manuscript. I look forward to reading your revised manuscript and the rebuttal letter.

Reviewer 1 ·

Basic reporting

This article presents original modelled data on the range of the lion (Panthera leo) from the late Pleistocene and Holocene and compares this against the modern distribution using ecological niche models and environmental stratification of palaeoclimate. The article is written clearly and in good English; however, there is significant textual restructuring that needs to be implemented and additional background to be written for the introduction. The authors scientific method appears to be scientifically sound; however, is difficult to follow, and this section often references methods from other papers. This needs significantly more work to clarify. The lack of structure in the methods section is reflected in the results section and needs further clarification. This lack of structure is also present in the discussion section. The conclusions clear.

Experimental design

The methods section is difficult to read and requires most information to be obtained from referenced papers. This section needs much greater clarification and transparency, which should be included within the main manuscript.

Validity of the findings

The findings appear valid; however, are not clearly structured and difficult to tell apart from referenced materials.

Additional comments

Major Comments
L50-118: In the introduction the authors give a well written overview of the background to some of the physical geographical barriers of the lion e.g. vegetation, and rivers, but need to introduce climatic parameters (the focus of the study) in much more detail. Please add a paragraph on climatic parameters of lion habitation with focus on enabling climatic conditions and environmental barriers.
L50-118: The authors also do not make any reference in the introduction to anthropogenic barriers (past or present). A small paragraph referencing the drivers/barriers to lion distribution as caused by people would benefit the introduction (even if dismissed as outside the scope of this study as a subsequent limitation). Please add a small paragraph on anthropogenic impacts to lions.
L127-136: This is part of the background section and not the methods. Please move to the introduction and incorporate into the suggested climatic parameters paragraph.
L137-145: This is a part of the background section; this is not a methodology. While this is a justification for the methodology, it needs to be put into the introduction section to set up the methods chosen.
L272-278: Descriptive and further justification of the methodology. This should be in the Introduction section or Methods section.
L280-291: This paragraph is highly speculative and should make further use of the modelled data generated. References in this paragraph also make the authors contribution and referenced articles difficult to disentangle. Please clarify.
L328-332: This is a key discussion point! Great! Please expand.
L442-448: Really nice and clear conclusions. Please adapt the discussion to reflect these.
L450: Perhaps change “Core findings” to “Key findings”?

Minor/Specific Comments
L27: Italicize “Pantera leo”
L105: Decapitalize “Vulnerable”
L106: Move reference Bauer et al., 2016 to the end of the sentence.
L154: Please define what the suitable bioclimatic conditions are.
L161: Unclear whether the data in table 1 is from Fei & Yu, (2016) or your own collected data. Please clarify the sentence.
L165: This sentence is not supported. Please reference the larger scale of lion home ranges.
Figure 7: Please add time periods to geological periods of time.
Figure 8: Detail illegible. Please split into several figures to make text legible.

Reviewer 2 ·

Basic reporting

No comments.

Experimental design

No comments.

Validity of the findings

No comments.

Additional comments

I have read the submitted manuscript entitled “A kingdom in decline: Holocene range contraction of the lion (Panthera leo) modelled with Global Environmental Stratification” by David Cooper and colleagues. I consider that the manuscript is scientifically interesting, important in terms of lion conservation, and well written.

Major comments
In terms of the contents I am concerned by the following five points.

Firstly, as the authors themselves appear to be fully aware it is very likely that lions were widely distributed across the current Sahara during the African humid period in the earlier Holocene (e.g. Yamaguchi et al. 2004, Barnett et al. 2006). Also, I see that the authors have made a substantial effort to overrule the results that the modelling suggested partially because the authors recognised the importance of the lions’ colonisation across the Sahara for their exodus from Africa to Eurasia. I consider the authors’ attempt to overrule the results appropriate to make more bioenvironmental sense concerning the recent colonisation history of the lion especially related to the African humid period (e.g. Adkins et al. 2006, Chen et al. 2020). However, then, it is difficult to avoid pointing out the possibility that the modelling that the authors developed may have major flaws. Such flaws may have resulted in incorrect suggestions (in addition to the Sahara case), which may have been overlooked simply because they are not as well-known as is the case of the Sahara. By any chance, isn’t it possible to make some adjustments to the models to do re-analyses?

Secondly, the similar concerns may apply to the authors’ modelling for the Last Glacial Maximum (LGM). Some modelling appears to suggest that the Sahara was covered by vegetation during the LGM more than it is today (e.g. Shao et al. 2018) consistent with what the authors’ modelling suggests. However, a substantially larger number of references suggests that the Sahara was hyper-arid during the LGM (e.g. Adkins et al. 2006, Ye et al. 2019) although there were some wetter periods during the last glacial period (e.g. Hoffmann et al. 2016). The authors’ modelling suggests results that strongly contradict the idea that the Sahara was hyper-arid during the LGM. Considering the authors’ modelling having suggested incorrect results for the African humid period, the authors may want to carefully re-evaluate their modelling for the LGM.

Thirdly, as the authors appear to acknowledge, there were lions during the Holocene in the southeast Europe substantially larger than the area that the authors appear to have used for the modelling, including the current Bulgaria, Romania, Ukraine, and Hungary (e.g. Daróczi-Szabó et al. 2020). Although Daróczi-Szabó et al. (2020) was published very recently, because the relevant data have been available for a while (e.g. Vörös et al. 1983) I wonder why the authors excluded those areas from the analysis. Is it possible to include those areas into the modelling?

Fourthly, it is biolgeographically very appropriate for the authors to restrict the area concerning lions’ colonisation because similar GIS analyses sometimes overlook the mobility of the focal organisms – e.g. a small mammals in Africa very unlikely colonise Australia even if modelling suggests the existence of very suitable habitats there. However, then, considering the mobility of lions (capable of dispersing > 200km within 1-2 years: e.g. Yamaguchi et al. 2004) and the large distribution range of the extinct cave lions (Panthera (leo) spelaea) throughout the northern Eurasia during the LGM (e.g. Barnett et al. 2009) the authors probably should have set the range of the lions’ possible colonisation substantially larger.

Fifthly, again as the authors themselves appear to be aware, partially due to the nature of the authors’ work concerning ecological niche of the lion, the large distribution of the extinct cave lions throughout the northern Eurasia during the LGM (e.g. Barnett et al. 2009) may cast a shadow over the modelling. It is possible that the extinct cave lion and the modern lion overlap their niches substantially if not completely. The authors state in the line 137 “In this study, we assume that the environmental niche of the lion has been stable”. It’s fine, but, the authors probably need to set the “environmental niche of the lion” much broader than that we estimate based on the geographical range of the modern lion recorded during the historical time. Based on this assumption, one may want to classify a substantial part of the northern Eurasia as potentially suitable habitat for the lion, and develop modelling accordingly. Based on the current modelling it appears that there are few suitable areas for lions to colonise in Eurasia even during the LGM, possibly highlighting the flaws of the modelling. I wonder if the authors may want to do modelling based on this assumption, and, at least, present the results in the Supplementary Materials for comparison.

Minor comments
Lines 181-182
Very briefly explain what those four variables are.

Line 184
Should “(-19°E, 94°W, - 36°S, 50°N)” be “(-19°W, 94°E, - 36°S, 50°N)”?



References
Adkins et al. (2006). The “African humid period” and the record of marine upwelling from excess 230Th in Ocean Drilling Program Hole 658C. Paleoceanography and Paleoclimatology. https://doi.org/10.1029/2005PA001200.

Barnett et al. (2006). The origin, current diversity, and future conservation of the modern lion (Panthera leo). Proceedings of the Royal Society B: Biological Sciences 273: 2119-2125.

Barnett et al. (2009). Phylogeography of lions (Panthera leo) reveals three distinct taxa and a late Pleistocene reduction in genetic diversity. Molecular Ecology 18: 1668-1677.

Chen et al. (2020). Feedbacks of soil properties on vegetation during the Green Sahara period. Quaternary Science Reviews. https://doi.org/10.1016/j.quascirev.2020.106389.

Daróczi-Szabó et al. (2020). Pending danger: Recent Copper Age lion (Panthera leo L., 1758) finds from Hungary. International Journal of Osteology. DOI: 10.1002/oa.2875.

Hoffmann et al. (2016). Timing and causes of North African wet phases during the last glacial period and implications for modern human migration. Scientific Reports. DOI: 10.1038/srep36367.

Shao et al. (2018). Statistical reconstruction of global vegetation for the last glacial maximum. Global and Planetary Change 168: 67-77.

Vörös et al. (1983). Lion remains from the Late Neolithic and Copper Age of the Carpathian basin. Folia Archaeologica Budapest 34: 33–50.

Ye et al. (2019). Multifunctionality debt in global drylands linked to past biome and climate. Global Change Biology. https://doi.org/10.1111/gcb.14631.

---

## Round 0.2 · accepted · Accept

Thank you for your resubmission to the previous major comments. I agree with the reviewers that you have made a fantastic job of responding to the corrects. There are a few minor typos but I believe these can be picked up during the proofing stage. As a result, I am happy to recommend that your paper be accepted.

Reviewer 1 ·

Basic reporting

No comment

Experimental design

No comment

Validity of the findings

No comment

Additional comments

The authors have carried out a good job in restructuring the article to flow more logically and have addressed most points raised in the previous round of reviews to satisfaction.

A few very minor points:

The authors need to check their referencing throughout as their automated reference manager appears to have malfunctioned leading to many errors. Please also check puctuation (missing full stops on L165; L185).

L630: The authors were asked to change “Core findings” to “Key findings” and stated that they had made this change. This change has not been made. Please make this change.

Otherwise, nice job in making the suggested amendments and well done on writing a great paper.

Best wishes,
William J Harvey

Reviewer 2 ·

Basic reporting

None.

Experimental design

None.

Validity of the findings

None.

Additional comments

I consider that the authors' replies to my comments and revised manuscript are appropriate. I have no further comments.